# Compositions and Biological Activities of Pomegranate Peel Polyphenols Extracted by Different Solvents

**DOI:** 10.3390/molecules27154796

**Published:** 2022-07-27

**Authors:** Yanlin Feng, Jiali Lin, Gang He, Li Liang, Qijun Liu, Jun Yan, Qian Yao

**Affiliations:** Key Laboratory of Medicinal and Edible Plants Resources Development of Sichuan Education Department, Sichuan Industrial Institute of Antibiotics, School of Pharmacy, Chengdu University, Chengdu 610106, China; fengyanlin19@163.com (Y.F.); linjiali202224@163.com (J.L.); cddxzyhx@hotmail.com (G.H.); liangli@cdu.edu.cn (L.L.); 17340176919@163.com (Q.L.)

**Keywords:** pomegranate peel, extraction solvents, polyphenols, antioxidant activity, antibacterial activity, anti-inflammatory activity

## Abstract

Pomegranate peel extract (PPE), which is abundant in polyphenols, holds immerse prospects for the treatment of airway infection. In this study, water and ethanol of 30%, 50%, and 80% were used to prepare PPE. A total of 18 phenols belonging to 8 categories of polyphenols were identified in PPE by HPLC-MS/MS. The PPE from the four extraction solvents possessed different antioxidant, antibacterial, and anti-inflammatory activities. Principal component analysis revealed that though total flavonoids (TFs), total polyphenols (TPs), and total tannins (TTs) were responsible for the reducing power of PPE, only TFs contributed to the effect of PPE in inhibiting lipid membrane peroxidation. TPs, TTs, and punicalagin were positively correlated with the antibacterial strength against *S. aureus* while TTs alone contributed to the inhibition of methicillin-resistant *S. aureus*, implying the crucial role of TT in suppressing bacteria. Meanwhile, TTs was associated with the prevention of IL-6 release. The PPE with higher contents of TPs, TTs, and punicalagin had a weaker capacity to decrease nitric oxide secretion. PPE of 30% ethanol gained the highest integrated score due to its stronger antioxidant, antibacterial, and anti-inflammatory activities. It is a suitable candidate for the therapy of respiratory tract infection.

## 1. Introduction

With the global outbreak of COVID-19, herbal medicines with a therapeutic effect on respiratory tract inflection have attracted growing attention. Respiratory tract infection is often accompanied by both virus and bacterial invasion. Virus infection prompts the load and spread of bacteria, such as *Staphylococcus aureus* (*S. aureus*) [1]. When infection occurs, the innate immune system triggered by the host recognizes the pathogens through pattern recognition receptors and stimulates the release of proinflammatory cytokines, leading to the assembly of inflammatory cells and the inflammatory response [2]. In the clinic, the use of antibiotics and anti-inflammatory drugs is often involved in the treatment of respiratory tract infection. Due to the severe problems of bacterial resistance, natural products with antibacterial and anti-inflammatory activities have become a promising alternative for the therapy of airway infection.

Pomegranate peel (PP) is a chief by-product from the manufacture of pomegranate juice. PP is abundant in polyphenols, flavonoids, and hydrolyzed tannins, which are responsible for its diverse biological activities [3]. PP is efficient in inhibiting various pathogens, including virus, Gram-positive and Gram-negative bacteria, fungi, and mold [4]. PP exhibits an excellent therapeutic effect on chronic inflammation, especially for digestive tract inflammation, such as ulcerative colitis [5]. In addition, PP has other health-care-benefitting properties, for example, reducing blood pressure, anti-diabetes, anticancer, etc. [6].

Owing to its multiple biological activities, PP shows broad opportunities in the remedy of respiratory tract infections. Different strategies have been used to recover the active components from PP. Kumar et al. [7] compared the efficiency of methanol, ethanol, water, acetone, and hexane in extracting major bioactive compounds such as ellagic acid and punicalagin from PP, and found that the use of the freeze-drying method and methanol as the extraction solvent was efficient in recovering more active components. Ranjha et al. [8] compared the effects of two extraction approaches, sonication and maceration, and three polar solvents, ethanol, methanol, and acetone, at the concentrations of 50% and 70%, on the total polyphenol (TP) content of PP. They concluded that sonication yielded higher TP than the maceration technique. The use of 50% methanol attained the highest TP of 72.21 mg gallic acid equivalent (GAE)/g from PP. Cam et al. [9] optimized the water extraction condition and obtained TP of 192 mg GAE/g of PP. They demonstrated that there was no statistical difference between water extraction and classical methanol extraction. Rajha et al. [10] used ultrasound to pretreat PP for 20 min, followed by hot water extraction at 50 °C. They obtained TP of 50 mg GAE/g. As the variety, cultivating surroundings, and climate have a fundamental impact on the quality of PP, it is difficult to estimate which method is more suitable for PP extraction. 

In this study, considering the low toxicity and environmental friendliness of ethanol, water and ethanol of different concentrations were used to extract PP under highly efficient ultrasonic treatment. The phenols in PPE were identified by HPLC-MS/MS. The contents of TP, flavonoids, tannins, punicalagin, and ellagic acid in pomegranate peel extracts (PPE) were determined, respectively. The antioxidant, antibacterial, and anti-inflammatory activities were assessed. The correlation between the biological activities and active ingredients, and the integrated scores of PPE yielded from different extraction solvents were processed by principal component analysis. The PPE with the highest score is a candidate for the treatment of respiratory tract infection.

## 2. Results and Discussion

### 2.1. HPLC-MS/MS Analysis

#### 2.1.1. Polyphenols

A total of 18 phenol compounds were identified in PPE from 30% ethanol (PPE-30) under negative mode (Table 1). The total ion flow chromatogram is displayed in Figure 1. Except compound **16** and **17**, the other 16 polyphenols also belonged to tannin. As the ionization degree is a dominant factor on the peak intensity, the responses of the components in the total ion flow chromatogram were different from the responses in the chromatogram detected by UV. Most compounds generated an [M − H]^−^ ion of *m*/*z* 179 or 181 in the MS^2^ experiment, which were assigned to glucose or glucitol. The fragment was derived from a monosaccharide, mannoheptulose monohydrate (*m*/*z* 227), or dehydrogenmannoheptulose monohydrate (*m*/*z* 225). They were decomposed into glucose or glucitol via the loss of water and carbon monoxide. It implies that the polyphenols of PPE are present in the form of glycosides.

The most abundant phenols in PPE are punicalagin and its isomers. Punicalagin contains the moieties of glucose, ellagic acid, and gallic acid [11]. The complex structure of punicalagin led to its diverse molecular compositions and different dissociation modes in MS. Compounds **2** and **7** were identified as punicalagin α and β, respectively, due to their consistent degraded fragments. The highest abundance ion of compound **2** and **7** was the ion with *m*/*z* 181 and 481, respectively, which was assigned to glucitol and ellagic acid-glucoside, respectively. The [M − H]^−^ of *m*/*z* 913 and 781 were derived from the parent molecule losing a gallic acid and ellagic acid moiety, respectively. The [M − 2H]^2−^ of *m*/*z* 541, which was equivalent to the half molecular mass of punicalagin, was the characteristic peak of punicalagin [12]. In addition, the MS^2^ shows that the ion at *m*/*z* 253 was decomposed into glucose by eliminating one carbon monoxide and two water molecules. 

The compounds **1**, **3**, and **5** were identified as punicalagin isomers, for they have [M − H]^−^ of *m*/*z* 1083. The MS^2^ shows that the parent molecules removed mannoheptulose monohydrate and formed the ion of *m*/*z* 856, or further dehydrogenated and yielded the ion at *m*/*z* 854. Meanwhile, the compounds **3** and **5** retained the characteristic peaks of punicalagin, such as *m*/*z* 781 and 481. The compound **12** was another punicalagin isomer with a different detachment manner. The fragments of ellagic acid at *m*/*z* 301 [13] and digalloylhexoside at *m*/*z* 483 [14] appeared in the MS of compound **12**. In MS^2^, the ion of *m*/*z* 913 was generated by losing gallic acid from the parent molecule.

The compounds **9**, **10**, and **18** were identified as granatin B and its isomers, for all of them had the molecular ion peak [M − 2H]^2−^ at *m*/*z* 951. The compound **9** and **10** had the ion of *m*/*z* 331, which was assigned to dehydrated galloyl-hexoide. Moreover, they had the characteristic ion of *m*/*z* 541, which belongs to the punicalagin family. The fragments of *m*/*z* 475 and 476 were assumed to be ellagic acid-deoxyhexoside [14]. The ions of *m*/*z* 783 and 784 in MS^2^ were assigned to the parent molecule losing one molecule of gallic acid. The fragment of compound **18** at *m*/*z* 603 in MS^2^ was from the parent molecule eliminating one molecule of glucitol. The ion of compound **10** with *m*/*z* 633 was assigned to galloyl-hexahydroxydiphenoyl-hexoside (galloyl-HHDP-hexoside) [12]. The fragment profile of compound **10** is in agreement with the results reported by Peršurić et al. [15] and was identified as granatin B.

#### 2.1.2. Flavonoids

A total of 11 flavonoids were identified in PPE-30 using positive ion mode [16,17] (Table 1b). The ions with *m*/*z* 183 and 127 appeared in the MS^2^ of most compounds. It is proposed that the backbone structure of flavonoid (Figure 2a) had a self-disproportionated reaction. Ring A and B were reduced to trihydroxybenzopyran (Figure 2b), which corresponded to the fragment with *m*/*z* 183. Accordingly, ring C was oxidized into pyrogallol (Figure 2c) and formed the fragment of *m*/*z* 127. 

Compounds **6**, **7**, and **11** with the molecular ion [M + H]^+^ of *m*/*z* 595 were identified as kaemperol-3-o-rutinoside [18]. They were decomposed in various routes. Compound **6** was dissociated into the ions of *m*/*z* 126 and 471, which further removed one molecule of kaemperol and generated the [M + H]^+^ of 186, the ion with highest abundance in the MS of compound **6**. The ion of *m*/*z* 200 from compound **7** was assigned to the reduced product of ring A and B of kamperol followed by hydroxylation. In the MS of compound **11**, the fragment of *m*/*z* 296 corresponded to rutinose. The ion of *m*/*z* 391 was phenol rutinoside.

The abundance of the fragment with the strongest response was set at 100%. The relative abundances of other fragments were obtained by the comparison with the highest response.

### 2.2. Yield and Active Ingredients

The results are displayed in Table 2. The PPE extracted with water and 80% ethanol, which is marked as PPE-W and PPE-80, had over a 40% yield while those with 30% (PPE-30) and 50% ethanol (PPE-50) attained yields from 30% to 35%. Total tannins (TTs) in PPE-W were significantly lower than in the other extracts (*p* < 0.05). This may be attributed to the higher solubility of tannins in alcohol. PPE-50 obtained the highest TP, TTs, and punicalagin. PPE-80 had the lowest total flavonoids (TFs) but with higher ellagic acid. Derakhshan et al. [19] used 80% ethanol to extract PP at room temperature for 48 h. The TP contents in the extracts from different varieties of PP ranged from 276 to 413 mg GAE/g, which is in accordance with our results. However, the TF contents of 36–54 mg rutin equivalent (RUE)/g were much lower than ours. Kazemi et al. [20] used 70% ethanol to extract PP under the assistance of probe ultrasound. The yield varied from 26.8% to 41.6%. The contents of TP, punicalagin, and ellagic acid fluctuated in the range of 272.05–320.26, 128.02–146.61, and 10.12–22.53 mg/g, respectively. The contents of TP and punicalagin in our study were generally higher, but ellagic acid was lower than what Kazemi et al. reported. Amini et al. [21] employed micro wet milling to treat PP and optimized the extraction parameters by response surface methodology. They selected 49.65% ethanol to extract PP and obtained PPE with TP of 225.7 mg GAE/g, punicalagin of 71.6 mg/g, and ellagic acid of 6.9 mg/g. The TP and punicalagin contents of our study were higher than the results acquired by Amini et al. Nevertheless, the ellagic acid contents were consistent with each other. The variation in the yield and active ingredient derives from the difference in the extraction solvents, methods, and PP varieties [3].

### 2.3. Antibacterial Activity

The inhibitory activity of PPE against S. aureus and methicillin-resistant S. aureus strains is shown in Figure 3. The aqueous extract exhibited the weakest activity against S. aureus while the other three PPE presented similar inhibitory strengths. The concentration required to suppress 50% bacteria (IC_50_) of PPE-W, PPE-30, PPE-50, and PPE-80 was 213, 201, 198, and 203 μg/mL, respectively. The aqueous extract also displayed poorer power in inhibiting methicillin-resistant S. aureus while PPE-30 and 80 showed more vigorous activity. The IC_50_ was 454, 369, 398, and 365 μg/mL, respectively. It seems that nearly two-fold PPE concentrations have to be used to suppress the bacteria resistant to antibiotics. The positive control of ampicillin presented markedly more vigorous strength in preventing the bacterial growth than PPE (*p* < 0.01).

Though the inhibitory effect of PPE is not comparable to antibiotic, PPE still presents moderate antibacterial activity. *S. aureus* was considered the most sensitive strain toward PPE [15]. In other studies, researchers found that PPE had an equivalent strength against *S. aureus*, *Salmonella* sp., *E. coli*, etc. [22]. The antibacterial activity of PPE was associated with its effect on impairing the membrane integrity of bacteria. Afterward, punicalagin could link with the functional domain of bacteria, leading to the regulation network being disordered and successive cellular damage [23].

### 2.4. Antioxidant Activity

Oxidant-antioxidant imbalance plays an important role in inflammation occurrence. A number of antioxidant reagents demonstrate promising anti-inflammatory effects [24]. The activity of PPE in scavenging DPPH and ABTS^+^ radicals is shown in Figure 4a,b. The PPE acquired by different extraction solvents presented a similar quenching capacity. The radical scavenging ability of ascorbic acid was significantly stronger than PPE (*p* < 0.05).

Ferric reducing antioxidant power (FRAP) of PPE is displayed in Figure 4c. The PPE has moderate FRAP. The concentrations of PPE-W, PPE-30, PPE-50, and PPE-80 required to yield an absorbance of 0.5 were 146.75, 125.59, 124.71, and 153.82 μg/mL, respectively. PPE-30 and 50 presented a significantly higher reducing power than those of PPE-W and PPE-80 (*p* < 0.05), which is due to the more plentiful TP and TF present in the PPE [25]. Ascorbic acid still displayed more powerful FRAP than PPE, with a value of 63.15 μg/mL. The reported scavenging capacity and FRAP of PPE varied significantly [8,26] due to the difference in the extraction and measurement methods, PP varieties, and units used to depict the scavenging rates.

The cell membrane consists of a number of polyunsaturated fatty acids containing phospholipids and cholesterol esters, which are the principal targets under oxidization stress [27]. The membrane can be readily oxidized through a free radical-induced lipid peroxidation process and form chemically reactive species, leading to the structural alteration of lipid bilayers, augmented membrane permeability, and chronic inflammation [28]. In this study, liposomes were used as the biological membrane model to assess the potential activity of PPE against lipid membrane peroxidization. The inhibitory effect of PPE is shown in Figure 4d. The effect order was PPE-30 > PPE-50 > PPE-W but with no significant difference (*p* > 0.05). The IC_50_ was 0.15, 0.16, and 0.18 μg/mL, respectively. The activity of PPE-80 was notably lower than other extracts, with an IC_50_ of 0.22 μg/mL (*p* < 0.05). The strength difference was correlated with the TF contents. The IC_50_ of tannin, the positive control, was 0.10 μg/mL, indicating the markedly stronger activity compared to PPE (*p* < 0.05). In animal experiments, methanol extract of PP was reported to substantially reduce the oxidation of erythrocyte membrane by lowering the malondialdehyde (MDA) level [29]. Our study also proposes the beneficial effect of PPE acquired by water and ethanol to hamper the peroxidation of the lipid membrane.

### 2.5. Anti-Inflammatory Activity

PPE of different concentrations had no detrimental effect on the RAW 264.7 cells. The viability of cells treated with PPE was maintained at over 98%. The cells secreted some proinflammatory cytokines under the stimulation of LPS. PPE can reduce cytokine secretion and display anti-inflammatory activity (Figure 5). The activity presented a dose-dependent mode. As the concentrations increased, the cytokine levels diminished gradually. PPE acquired from different solvents exhibited a potent effect in preventing IL-6 release. The strength of PPE-W, PPE-50, and PPE-80 was similar to each other, with an IC_50_ of 100, 95, and 98 pg/mL, respectively. PPE-30 possessed much stronger activity, with an IC_50_ of 87 pg/mL. The effect of PPE in reducing the nitric oxide (NO) level was not so prominent. Even though the PPE concentrations were as high as 200 μg/mL, the NO levels were only reduced to 27.64%, 25.63%, 22.11%, and 26.13% by PPE-W, PPE-30, PPE-50, and PPE-80, respectively. PPE had the capacity to diminish PGE2 secretion as well, lowering the levels by 46.87%, 59.49%, 42.67%, and 36.25% when the dose was 100 μg/mL, respectively.

Du et al. [30] employed 60% ethanol to perform ultrasound-assisted extraction of PP and examined the anti-inflammatory effect on RAW264.7 cells. They found that with the treatment of 100 μg/mL PPE, the levels of IL-6, NO, and PGE2 were reduced by 66%, 80%, and 35%, respectively. The effect against PGE2 was lower, but the strength against NO was more potent with respect to the results of our study. The difference in the cellular density and the LPS dose used may contribute to the variation in the result. In other cellular models, PPE reduced proinflammatory cytokines’ expression in BME-UV1 cells, a bovine mammary epithelial cell line [31].

### 2.6. Principal Composition Analysis

#### 2.6.1. Correlation Analysis

The correlation between the activity and composition of PPE, which was analyzed by SPSS software, is displayed in Table 3. Ellagic acid is negatively correlated with most activities, for example, antibacterial and inhibitory effects against cytokine secretion. Ellagic acid was reported to have capacity in suppressing bacterial growth, including Gram-positive and Gram-negative bacteria and fugus [32]. The scavenging capacity of ellagic acid against DPPH radicals was even stronger than quercetin [33]. However, in our study, it seems that ellagic acid had a negative influence on the extract activities. The possible reason is that ellagic acid has four phenolic hydroxyls and two carbonyls, which are readily complexed with other compounds via the interaction between hydroxyls and carbonyls. Since phenolic hydroxyls and carbonyls play a crucial role in the antioxidant and antibacterial process of polyphenols and flavonoids, such interplay alleviates the strength of polyphenols and flavonoids.

TT is primarily responsible for antibacterial activity of PPE. The inhibitory effect of PPE toward both *S. aureus* and methicillin-resistant *S. aureus* increased with the elevation of the TT concentration. However, TT exerted a contrary impact on suppressing NO secretion. The PPE with higher TT had poorer activity in inhibiting NO. Not only TT, TP also presented the same trend. In Table 4, it shows that the PPE with more potent power in preventing the production of IL-6 and PGE2 will have a weaker strength in decreasing NO. Meanwhile, the effect of suppressing NO is inversely correlated with FRAP and its inhibitory effect on lipid membrane oxidation. NO is a surrogate marker for inflammation, as large amounts of NO are released upon inflammation [34]. On the other hand, NO is a cellular antioxidant, capable of breaking free radical-mediated lipid peroxidation [35]. Microsomal lipid peroxidation was decreased in a dose-dependent manner by exposure to NO [36]. In the RAW264.7 cellular experiments, the four PPE exhibited anti-inflammatory properties by reducing NO levels in a concentration-dependent mode. However, when the four PPE were maintained at the same concentration, it was found that the inhibitory effect against NO was in contrast with the contents of TT and TP in PPE. The cells incubated with the extract that contains higher TT and TP will secrete more NO, which further hinders the lipid membrane from being oxidized. In another word, PPE can suppress NO release in inflammatory surroundings. The high contents of TT and TP in PPE will lead to a weak inhibitory effect toward NO and produce relatively higher levels of NO, which in turn facilitates the antioxidant process.

Apart from TT, TP and punicalagin are also positively related with the antibacterial activity against *S. aureus*, but they have no contribution to the inhibition of methicillin-resistant *S. aureus*. Only TT presents the positive correlation with the inhibitory effect on methicillin-resistant *S. aureus*., implying the fundamental role of TT in antibacterial strength. TF contributes most to FRAP and inhibition of lipid peroxidation while TP is primarily related with FRAP. Table 4 displays the close relationship between anti-inflammation and antioxidant activity, which have already been verified by many studies [37].

#### 2.6.2. Principal Component Analysis

Based on the contributions to the total variance, three principal components (PC) were selected to represent the original twelve variables. PC1, PC2, and PC3 explained up to 55.00%, 27.22%, and 17.78% of the total variance, respectively. The loading relationship between the twelve variables and the three PC is shown in Figure 6. It indicates that PC2 is closely related with TF, the inhibitory rate against PGE2, and methicillin-resistant *S. aureus*. PC3 is correlated with punicalagin. The other nine variables were loaded onto PC1 with high correlation (*r* > 0.65).

The integrated scores of PPE-W, PPE-30, PPE-50, and PPE-80 were −185.03, 138.25, 110.09, and −63.31, respectively. PPE-W obtained the lowest score due to the poor extraction ability of water for tannins, which is a *Crit.* component in inhibiting bacteria. PPE-30 attained the highest score owing to its strong activity in resisting oxidation, bacterial growth, and the formation of proinflammatory cytokines.

## 3. Materials and Methods

### 3.1. Materials

Pomegranate peel (PP) was obtained from Tongrentang Drug Store (Bozhou, Anhui province, China), and was dried at 40 °C for 12 h. The punicalagin reference with a purity ≥ 98% (determined by HPLC) was obtained from the National Institute of Pharmaceutical and Biological Products (Beijing, China). 2,2-Diphenyl-1-picrylhydrazyl (DPPH) was purchased from Kelong Chemical Reagent Company (Chengdu, China). Liposomes were prepared by our lab. The size of the liposomes was 121.3 ± 3.8 nm, as measured by a Zen 3600 nanoparticle sizer (Malvern Instrument Co., Ltd., Malvern, UK). Dulbecco’s modified Eagle medium (DMEM), fetal bovine serum (FBS), penicillin, and streptomycin were acquired from Thermo Fisher Scientific Co., Ltd. (Shanghai, China). Lipopolysaccharide (LPS) from *Escherichia coli O55:B5* was obtained from Sigma-Aldrich (St. Louis, MO, USA). 2,2′-azino-bis(3-ethylbenzothiazoline-6-sulfonic acid) (ABTS) and 3-(4,5-Dimethyl-2-thiazolyl)-2,5-diphenyl-2H-tetrazolium bromide (MTT) were purchased from Shanghai Macklin Biochemical Co., Ltd. (Shanghai, China).

### 3.2. Extraction of PP

The PP were ground into powders, passed through an 80-mesh sieve, and stored under −20 °C until use. The particle size was 512.6 ± 14.8 μm, which was determined by a Bettersize2600 particle size instrument (Better Instrument Co., Ltd., Dandong, Liaoning province, China). The moisture in the powders, assayed by a YLS16A(pro) moisture instrument (Shanghai Tianmei Balance Instrument Company, Shanghai, China), was 4.57 ± 0.74% (*w*/*w*). Four solvents were employed to extract PP with the liquid/material ratio of 25:1: distilled water, 30% ethanol, 50% ethanol, and 80% ethanol (*v*/*v*). Ultrasound-assisted extraction was conducted according to the following conditions: temperature 50 °C, ultrasound power 240 W, and extraction time 60 min. After the extraction was completed, the solution was filtered and concentrated to 5 mL by an RE-2000A rotary evaporator (Shanghai Yarong Biochemical Instrument Factory, Shanghai, China). Then, the remnant solution was freeze dried to a constant weight by a LABConco freeze drier (Guangzhou Saituotong Technology Co., Ltd., Guangzhou, China). The yield was calculated by comparison with the initial weight of PP. The dried pomegranate peel extract (PPE) was kept at −20 °C for further analysis.

### 3.3. Analysis of the Extracts

#### 3.3.1. Composition Identification by HPLC-MS/MS

PPE-30 was prepared to 1 mg/mL, from which 2 μL was injected into a Waters Acquity UPLC system (Waters Corporation, Milford, MA, USA). The isolation was carried out on an Acquity UPLC HSS T3 column (2.1 × 150 mm, 1.8 μm). The column temperature was maintained at 40 °C. For the detection in negative ion mode, the mobile phase consisted of acetonitrile (A) and 5 mM ammonium formate solution (B). Gradient elution was conducted as follows: phase A maintained 2% from 0 to 1 min and was elevated to 50% from 1 to 9 min. Then, it continuously rose to 98% in 3 min and was held for 1.5 min. Afterward, the ratio dropped to 2% in 0.5 min and was held for 3 min. For the positive ion mode, the mobile phase was composed of 0.1% formic acid in acetonitrile (C) and 0.1% formic acid aqueous solution (D). Phase C kept 2% from 0 to 9 min, and rose to 50% from 9 to 12 min. Then, phase C elevated to 98% in 0.5 min, held for 0.5 min, and returned to 2% from 14 to 20 min. 

MS analysis was performed by a Thermo Q Extractive MS Detector (Thermo Fisher Scientific, Waltham, MA, USA). Electro-spray ionization (ESI) was used in both negative and positive ionization modes. The ion source parameters were set as follows: spray voltage of −2.50 kv for negative mode and 3.5 kv for positive mode, sheath gas 30 arb, auxiliary gas 10 arb, and capillary column temperature of 350 °C. The primary full scan was performed from *m*/*z* 100 to 2000 at a high resolution of 70,000. The top 10 ions with strong signals performed secondary breaking using HCD with a collision energy of 30 ev and resolution of 17,500. The identification of the compounds was conducted by the analysis of MS fragments and comparison with the references.

#### 3.3.2. Total Polyphenols 

The determination of total polyphenols (TPs) was carried out using Folin–Ciocalteu reagent and following the methods reported by Lu et al. [38]. The results were expressed as mg GAE/g.

#### 3.3.3. Total Flavonoids 

The contents of total flavonoids (TFs) in the extracts were determined according to what Papaioannou et al. [39] proposed and were denoted as mg RUE/g.

#### 3.3.4. Total Tannins 

The determination of total tannins (TTs) conformed to what Lima et al. [40] reported with some modification. Briefly, PPE was prepared at 1 mg/mL with the corresponding extraction solvents, respectively, from which 1 mL was drawn out, mixed with 1 mL of ferrous tartrate reagent (ferrous sulfate heptahydrate of 1 g, potassium sodium tartrate of 2 g, and sodium bisulfite of 0.1 g were dissolved in 100 mL of distilled water), and reacted at room temperature (22–25 °C) for 5 min. Afterward, 4 mL of water was added and the absorbance at 560 nm was determined. The TT contents in the extracts were calculated according to the absorbance of tannin standard solutions.

#### 3.3.5. Ellagic Acid and Punicalagin 

The extracts were prepared at 1 mg/mL with water and were isolated on an Elite C18 column (4.6 mm × 250 mm, 5 µm), respectively. The analysis was conducted by an Agilent 1100 HPLC instrument (Agilent Technologies, PaloAlto, CA, USA). The mobile phase consisted of methanol (A) and 0.2% phosphoric acid (B) with gradient elution. Phase A rose from 5% to 36% in 15 min, and elevated to 60% from 15 to 20 min. Afterward, it began to drop to 5% from 20 to 23 min and held for 3 min. The flow rate was 1 mL/min. The column temperature was maintained at 35 °C. The injection volume was 10 µL.

### 3.4. Antibacterial Activity 

*Staphylococcus aureus* (*S. aureus*, ATCC 25923) and methicillin-resistant *S. aureus* (ATCC 33591) were provided by National Strain Preservation Center, Sichuan Industrial Institute of Antibiotics (Chengdu, China). The bacteria were cultured in LB medium overnight. Then, the bacterial density was adjusted to 10^6^ CFU/mL, from which 3 mL was taken out, mixed with 2 mL of PPE solutions, and cultured under 37 °C in a shaking bed with a shaking speed of 100 rpm for 8 h. The turbidity at 600 nm (*A*) was measured by an ELX800 Microplate Reader (Agilent Technologies, PaloAlto, CA, USA). In addition, 2 mL of distilled water in the place of the sample was manipulated as mentioned above and the turbidity was denoted as *A*_0_. Ampicillin of 0.125 mg/mL was set as the positive control. The inhibitory rate was calculated based on the following Equation (1). The antibacterial curve was plotted using the inhibitory rate versus the PPE concentration and the concentration to inhibit 50% cell growth was estimated from the curve:Inhibitory rate = (*A*_0_ − *A*)/*A*_0_ × 100%(1)

### 3.5. Antioxidant Activity

#### 3.5.1. Scavenging ABTS^+^ Radicals

The scavenging activity was determined following the method described by Re et al. [41]. Ascorbic acid was set as the positive control.

#### 3.5.2. Scavenging DPPH Radicals

The scavenging activity was determined according to the method described by Yao et al. [42] with slight modification. The sample of 1.0 mL was mixed with 3.0 mL of 0.1 mM DPPH solution and stood at room temperature in the darkness for 30 min. The absorbance at 517 nm was recorded.

#### 3.5.3. Ferric-Reducing Antioxidant Power

The ferric-reducing antioxidant power (FRAP) was determined according to what we proposed before [42].

#### 3.5.4. Inhibition against Lipid Peroxidation

The sample of 0.5 mL was blended with 0.5 mL of liposomes and incubated under 37 °C for 60 min. Subsequently, 1 mL of 1% thiobarbituric acid (*w*/*v*) was added and boiled for 10 min to develop color. After the solution was cooled to room temperature, the absorbance at 532 nm (*A*) was measured, Meanwhile, 0.5 mL of water replaced the sample and the measured absorbance was marked as *A*_0_. The inhibitory rate against lipid peroxidation was calculated according to Equation (1). Tannin was set as the positive control.

### 3.6. Anti-Inflammatory Effect

#### 3.6.1. Cell Culture

The mouse macrophage RAW264.7 cell line was purchased from National Collection of Authenticated Cell Cultures (Shanghai, China). The cells were grown in DMEM supplemented with 10% FBS, 100 U/mL penicillin, and 100 μg/mL streptomycin. They were propagated in a 37 °C incubator infused with 5% CO_2_. Cells at a density of 4 × 10^4^/well were seeded onto a 96-well plate and were cultured overnight. Afterward, the mediums were replaced with 200 μL of fresh ones containing different concentrations of PPE, and incubated under 37 °C for 2 h, respectively. Then, 20 μL LPS of 1 μg/mL was added and cultured for another 24 h.

#### 3.6.2. Cell Viability

After the cells were co-cultured with PPE for 24 h, 20 μL MTT of 5 mg/mL was added and incubated under 37 °C for 4 h. Then, the supernatant was discarded and 150 μL DMSO was added to dissolve the crystals. The absorbance of 570 nm was measured. The cells co-cultured with distilled water were set as the control and its cellular viability was set at 100%.

#### 3.6.3. Determination of IL-6 and PEG2

The levels of IL-6 and PEG2 in the supernatants of culture mediums were measured following the manufacturer’s instructions regarding ELISA kits (MultiSciences Lianke Biotech Co., Ltd., Wuhan, China).

#### 3.6.4. Determination of Nitric Oxide

In total, 80 μL of supernatant was taken out from the culture medium, mixed with 80 μL of Griess reagent A (0.5 g of p-aminobenzene sulfonic acid was dissolved in 150 mL of 10% acetic acid), and incubated at 30 °C in the dark for 10 min. Afterward, 80 μL of Griess reagent B (0.1 g phenylamine was dissolved in the solvent, including 20 mL distilled water and 100 mL of 10% acetic acid) was added and reacted at 30 °C for another 10 min. The absorbance at 540 nm was measured by a microplate reader. In addition, the mixture consisting of Griess reagent and culture medium was set as the blank control. Sodium nitrite solutions of different concentrations were used as the standards to plot the standard curve for the quantification [43].

### 3.7. Statistical Analysis

All data were averages of triplicate experiments and were expressed as the mean ± standard derivation (SD). Statistical analyses were performed using IBM SPSS Statistics 25 software (Chicago, IL, USA). The difference was considered statistically significant when *p* < 0.05.

Principal component analysis was conducted to assess the PPE in an integrated mode. The data, including the contents of active ingredients, FRAP, inhibitory rates of PPE against lipid peroxidation, bacterial growth, and pre-inflammatory cytokines, were input into SPSS software and normalization and dimension reduction processing were performed. Based on the contributions to the total variance, PC that were able to represent the original variables were determined. The integrated scores of PPE were calculated according to the following Formula (2):Total score = a_1_ × b_1_ × FAC1 + a_2_ × b_2_ × FAC2 + a_3_ × b_3_ × FAC3(2)
where a_1_ to a_3_ represent the square root of the characteristic value of PC, respectively. The coefficients b_1_ to b_3_ represent the contribution of PC to the total variance, respectively. FAC1 to FAC3 stand for the attribute vectors of PC, which were obtained by software, respectively.

## 4. Conclusions

The compositions of PPE were identified by HPLC-MS/MS. A total of 18 polyphenols and 11 flavonoids, including 5, 3, and 3 isomers of punicalagin, granatin B, and castalagin, were detected, respectively. The results from principal component analysis indicate that TF is responsible for both the FRAP and resistance against lipid peroxidation. TP and TT are positively related with FRAP but showed no contribution to the inhibitory effect on lipid peroxidation. Though TP, TT, and punicalagin contribute to the antibacterial activity of PPE against *S. aureus*, only TT plays a fundamental role in inhibiting methicillin-resistant *S. aureus*. In addition, TT has a weak positive correlation with the inhibition of IL-6. The PPE with higher contents of TP, TT, and punicalagin has poorer strength in preventing NO release. PPE-30 acquired the highest score due to the stronger antioxidant, antibacterial, and anti-inflammatory activities. It holds immerse prospective in the treatment of respiratory tract inflammation.

## Figures and Tables

**Figure 1 molecules-27-04796-f001:**
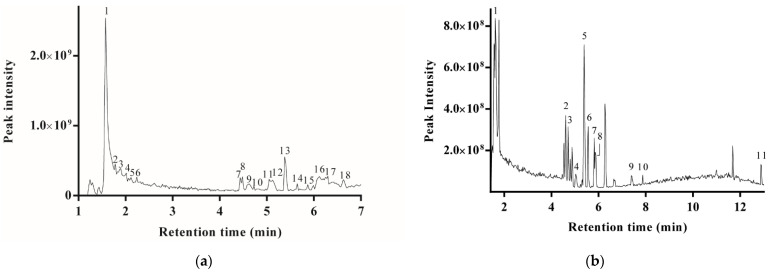
The total ion flow chromatogram detected by MS in negative ion mode (**a**) and positive ion mode (**b**) for the identification of polyphenols and flavonoids, respectively. The peak identifications are shown in Table 1a,b, respectively.

**Figure 2 molecules-27-04796-f002:**
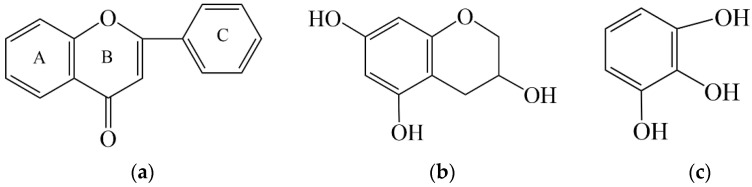
The backbone structure of flavonoid (**a**), reduced product of ring A and B corresponding to [M + H]^+^ of *m*/*z* 183 (**b**), and the oxidized product of ring C corresponding to [M + H]^+^ of *m*/*z* 127 (**c**) after MS collision.

**Figure 3 molecules-27-04796-f003:**
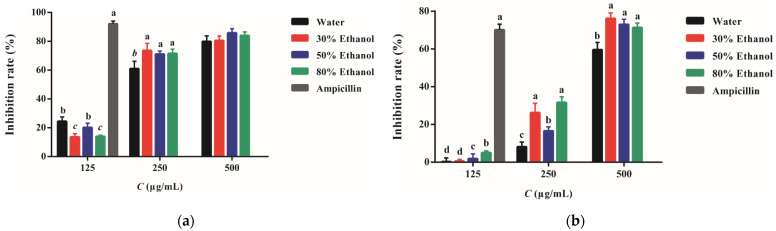
The inhibitory activity of PPE against *S. aureus* (**a**) and methicillin-resistant *S. aureus* strains (**b**). Different letters represent a statistical difference among the PPE (*p* < 0.05).

**Figure 4 molecules-27-04796-f004:**
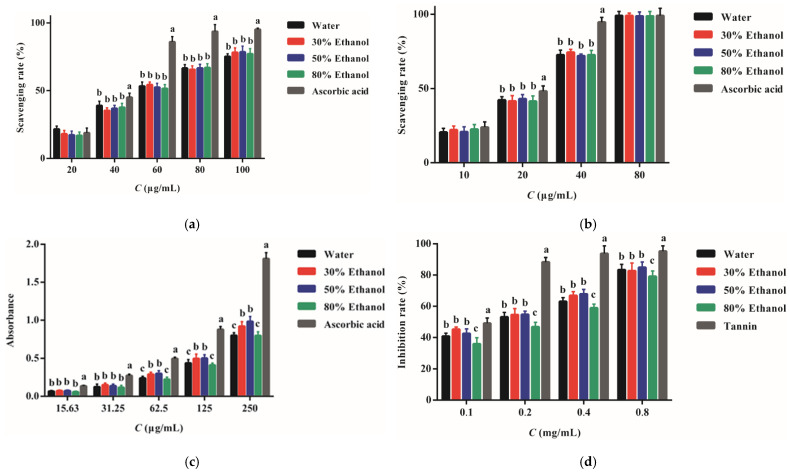
The antioxidant activity of PPE: scavenging DPPH radicals (**a**) and ABTS^+^ radicals (**b**), ferric reducing antioxidant power (FRAP) (**c**) and the inhibitory effect against lipid membrane peroxidization (**d**). Different letters represent statistical difference among the PPE (*p* < 0.05).

**Figure 5 molecules-27-04796-f005:**
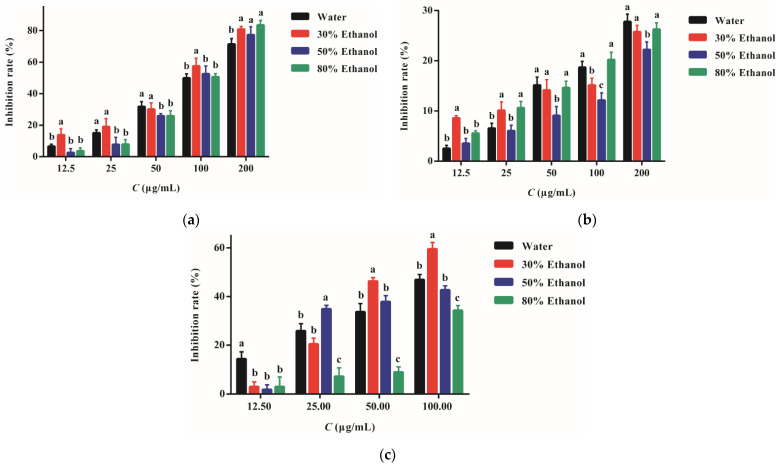
The inhibitory effect of PPE from different extraction solvents on the secretion of IL-6 (**a**), NO (**b**), and PGE2 (**c**). The measurement was performed on RAW264.7 cells pretreated with LPS of 1 μg/mL. Different letters represent a statistical difference among the PPE (*p* < 0.05).

**Figure 6 molecules-27-04796-f006:**
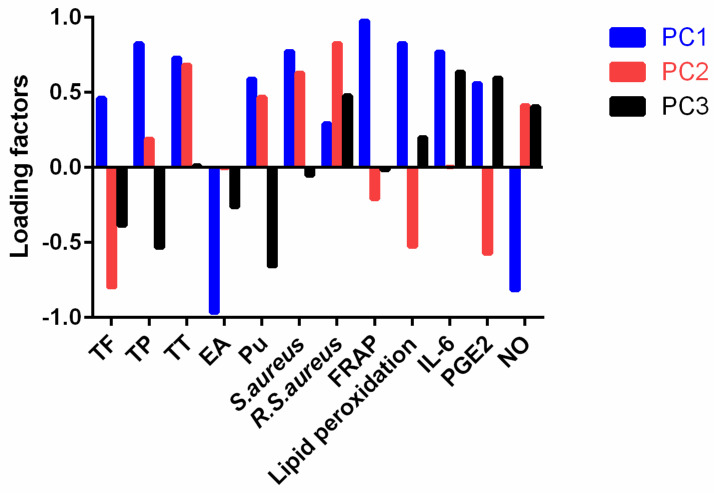
Loading plot of the twelve variables onto the three PC by principal component analysis. EA: ellagic acid; Pu: pullicalagin; *R. S. aureus*: Methicillin-resistant *S. aureus;* PC: principal component.

**Table 1 molecules-27-04796-t001:** (**a**) Identification of polyphenols in PPE yielded by 30% ethanol using HPLC-MS/MS in negative ion mode. (**b**) Identification of flavonoids in PPE yielded by 30% ethanol using HPLC-MS/MS in positive ion mode.

**(a)**						
**NO.**	**RT** **(min)**	**Name**	**Molecular** **Formula**	**[M − H]^−^/** **[M − 2H]^2−^**	**MS Fragment** **(*m*/*z*)**	**Relative** **Abundance (%)**
1	1.56	Punicalagin isomer	C_48_H_28_O_30_	1083.0603	856.4002	0.02
					227.0704	14.40
					181.0705	100.00

2	1.78	Punicalagin α	C_48_H_28_O_30_	1083.0603	913.1799	0.18
					781.0536	5.87
					481.0626	0.07
					253.0577	0.65
					179.0551	100.00

3	1.90	Punicalagin isomer	C_48_H_28_O_30_	1083.0603	854.1169	0.02
					781.0535	0.38
					481.0626	3.85
					225.0608	33.33
					179.0551	100.00

4	2.01	Punicalin	C_34_H_22_O_22_	781.0539	481.0624	2.09
					225.0607	34.38
					179.0551	100.00

5	2.11	Punicalagin isomer	C_48_H_28_O_30_	1083.0603	854.9579	0.02
					781.0524	0.04
					651.0829	0.05
					481.0623	2.12
					225.0607	34.62
					179.0551	100.00

6	2.24	Castalagin isomer	C_41_H_26_O_26_	932.4875	876.4804	0.03
					781.0540	0.21
					481.0624	2.46
					225.0608	33.46
					179.0551	100.00

7	4.44	Punicalagin β	C_48_H_28_O_30_	1083.0603	913.1799	1.15
					781.0536	11.15
					481.0624	100.00
					253.0921	1.50
					179.0551	2.46

8	4.48	Castalagin isomer	C_41_H_26_O_26_	933.0618	331.0667	100.00
					225.0608	10.00
					179.0551	30.74

9	4.63	Granatin B isomer	C_41_H_28_O_27_	951.0779	783.0662	0.38
					541.0249	100.00
					475.0331	0.30
					331.0667	38.75
					225.0608	13.13
					179.0552	41.25

10	4.73	Granatin B	C_41_H_28_O_27_	951.0706	633.0741	6.07
					541.0249	33.57
					475.0331	67.14
					331.0667	100.00
					225.0608	14.29
					181.0706	45.00

11	5.05	Castalagin	C_41_H_26_O_26_	933.0612	541.0248	100.00
					483.0777	16.67
					225.0608	9.58
					181.0705	30.83

12	5.12	Punicalagin isomer	C_48_H_28_O_30_	1083.0603	913.1811	0.59
					541.0250	100.00
					483.0778	2.82
					301.0562	2.09
					226.0648	11.36
					179.0552	33.64

13	5.38	Punigluconin	C_34_H_26_O_23_	801.0774	611.1405	14.36
					463.0523	5.27
					305.0664	100.00
					225.0608	4.91
					181.0705	12.91

14	5.65	Granatin A isomer	C_41_H_28_O_27_	783.0663	632.0633	13.53
					483.0779	4.47
					291.0141	100.00
					181.0706	47.65

15	5.87	Granatin A	C_41_H_28_O_27_	783.0663	447.0571	29.38
					305.0664	100.00
					225.0608	13.13
					181.0705	50.00

16	6.12	Ellagic acid	C_14_H_6_O_8_	300.9991	300.9991	100.00
					225.0608	8.21
					181.0705	39.29

17	6.30	Galloyl-HHDP-hexoside	C_27_H_22_O_18_	633.0739	392.5591	6.79
					352.0667	12.14
					325.0929	15.36
					300.9991	71.43
					289.0710	100.00
					181.0705	35.71

18	6.63	Granatin B isomer	C_41_H_28_O_27_	951.0705	784.0748	0.91
					603.0052	7.39
					476.0417	2.30
					300.9991	100.00
					225.0608	10.43
					181.0705	52.17
**(b)**						
**NO.**	**RT** **(min)**	**Name**	**Molecular** **Formula**	**[M + H]^+^**	**MS Fragment** **(*m*/*z*** **)**	**Relative** **Abundance (%)**
1	1.62	Syringetin hexoside	C_23_H_22_O_13_	507.1923	398.1655	4.40
					325.1123	100.00
					183.0863	42.86
					165.0762	21.43
					145.0496	29.76
					127.0391	35.71

2	4.61	Hesperidin	C_28_H_34_O_15_	611.1398	464.1777	0.78
					315.0718	21.62
					274.0985	0.30
					183.0863	6.49
					144.1384	100.00

3	4.71	Hesperidin isomer	C_28_H_34_O_15_	611.1388	436.1611	0.27
					315.0715	9.03
					248.1131	1.42
					183.0863	8.71
					124.1123	100.00

4	5.03	Rutin	C_27_H_30_O_16_	611.1385	611.1385	100.00
					448.0601	0.11
					305.0652	15.12
					172.1333	37.21
					127.0392	19.77

5	5.39	Gallocatechol	C_15_H_14_O_7_	307.0807	307.0807	100.00
					183.0863	4.51
					139.0389	6.62

6	5.56	Kaempferol-3-*O*-rutinoside	C_27_H_30_O_15_	595.1460	471.0188	5.62
					367.1505	1.75
					186.1492	100.00
					126.1279	4.69

7	5.82	Kaempferol-3-*O*-rutinosde	C_27_H_30_O_15_	595.1459	467.0822	1.85
					315.0715	1.58
					265.1440	1.42
					200.0470	100.00

8	5.86	Gallocatechol	C_15_H_14_O_7_	307.0807	307.0807	100.00
					183.0863	8.95
					127.0392	12.63

9	7.41	Quercetin	C_15_H_10_O_7_	303.0135	303.0135	100.00
					183.0864	43.42
					127.0391	19.74

10	7.87	Kaempferol-7-*O*-glucosie	C_21_H_20_O_11_	449.1079	449.1079	100.00
					371.2064	8.05
					287.0555	21.95
					183.0864	52.44

11	12.88	Kaempferol-3-*O*-rutinoside	C_27_H_30_O_15_	595.1459	468.4403	5.46
					391.2841	0.37
					296.2247	100.00
					183.0864	27.69

The highest abundance of the fragment was set at 100%. The relative abundance of other fragments was obtained by the comparison with the highest response.

**Table 2 molecules-27-04796-t002:** Yield and active components of PPE acquired by different solvents.

Active Ingredients	Extracts			
	Water	30% Ethanol	50% Ethanol	80% Ethanol
Yield (mg/g)	425.40 ± 26.01 ^a^	299.50 ± 22.11 ^b^	346.50 ± 19.17 ^b^	466.60 ± 26.04 ^a^
TP (mg GAE/g)	333.90 ± 29.59	340.60 ± 19.18	353.29 ± 26.91	335.63 ± 24.10
TF (mg RUE/g)	155.95 ± 14.18 ^a^	146.45 ± 17.69 ^a^	156.29 ± 15.70 ^a^	125.33 ± 16.61 ^b^
TT (mg tannin/g)	245.04 ± 16.50 ^b^	345.99 ± 17.18 ^a^	362.22 ± 19.71 ^a^	329.04 ± 20.18 ^a^
Ellagic acid (mg/g)	6.77 ± 0.69 ^a^	4.35 ± 1.50 ^b^	5.51 ± 0.61 ^b^	7.55 ± 0.48 ^a^
Punicalagin (mg/g)	112.74 ± 9.15 ^c^	123.92 ± 10.80 ^b^	165.85 ± 9.95 ^a^	132.00 ± 12.90 ^b^

TPs: total polyphenols; TFs: total flavonoids; TTs: total tannins; GAE: gallic acid equivalent; RUE: rutin equivalent. Different letters represent a statistical difference among the PPE yielded by various extraction solvents (*p* < 0.05).

**Table 3 molecules-27-04796-t003:** Correlation between activity and compositions.

Inhibitory Activity or FRAP		Correlation Coefficient			
	TF	TP	TT	Ellagic Acid	Punicalagin
*S. aureus*		0.786	0.995	−0.735	0.787
Resistant *S. aureus*	−0.711		0.786		
FRAP	0.626	0.776	0.569	−0.938	
Lipid peroxidation	0.724			−0.849	
IL-6			0.573	−0.910	
PGE2				−0.696	
NO		−0.981	−0.804	0.680	−0.554

FRAP: ferric-reducing antioxidant power. The contents of active ingredients in the four PPE and the corresponding FRAP, inhibitory rates against lipid peroxidation, bacterial growth, and cytokine secretion were input into IBM SPSS Statistics 25 software, respectively. The correlation between the activity and the component content was calculated by the software.

**Table 4 molecules-27-04796-t004:** Correlation coefficients between anti-inflammatory and antioxidant activity.

Antioxidant Activity	Anti-Inflammation		
	IL-6	PGE2	NO
FRAP	0.740	0.659	−0.891
Anti-lipid peroxidation	0.762	0.886	−0.809

## Data Availability

Data are included within the article.

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
