# Peer review of "Compositions and Biological Activities of Pomegranate Peel Polyphenols Extracted by Different Solvents"

_molecules, 2022, doi:10.3390/molecules27154796_

Round 1

Reviewer 1 Report

Dear authors,

after reviewing your paper "Composition and Biological Activities of Pomegranate Peel Polyphenols Extracted by Different Solvents", I concluded that the research you have conducted and the methods used are in accordance with modern scientific research. The research is very well designed, and the way of writing, presentation of results and their interpretation are at a high scientific level. Therefore, I proposed to the editors of the journal Molecules that this article be published in this form.

With respect,

reviewer

Author Response

Dear reviewer,

Thank you so much for your favorable comments on our manuscript. It gives us great pleasure and encouraging. We will keep working hard to prompt the development of study in our field.

Thank you again for your great efforts and time.

Yours faithfully,

Jun Yan

Qian Yao

Reviewer 2 Report

The authors in the current study investigated the (Compositions and Biological Activities of Pomegranate Peel Polyphenols Extracted by Different Solvents)The paper is very interesting specially in correlating the biological activities to the polyphenols content of the extracts. The following points should be clarified by the authors: • The paper is misleading for the reader and very confusing. • In abstract, material and methods, results and discussion all showed the effect or the content of flavonoids although no flavonoid components were identified or mentioned in the results. • Table1 which identify the polyphenol content of 30% extract contains no flavonoids !!. • The authors should investigate the polyphenols content by HPLC-MS/MS for all different extracts of water and ethanol of 30%, 50%, and 80% (not only 30%). As this study is a comparative study. • In PPC the correlation between the components and the activity should be clarified. • Table 3. Correlation between activity and compositions. Very confusing. Is it for 30% ONLY?? And if WHY?? The authors should give a clear legend for this table • PEE mentioned through the text is very misleading. The extract itself should be given 30 or 50 or…….

Reviewer 3 Report

The comments are as follows: 

1. I suggest putting the reference number after Author's name i.e., Kumar et al. [9]. Please, revise it through the text.

2. Please, provide the moisture content and particle size of the sample material. 

3. To make the results clearer and relevant, I suggest including the m/z experimental, m/z calculated and (M-H)- molecular formula columns in Table 1. As well, m/z values should be with four decimal places and the fragments with relative abundance.

4. Please, revise the text for some misprints. 

Round 2

Reviewer 2 Report

line 125: Total 11 flavonoids were identified in PPE. Which PPE? 30%, or……  clarify

line 125: Total 11 flavonoids were identified in PPE using positive ion mode.  Flavonoids showed clear and intense peaks in negative mode.  Why in this study in the positive mode?

Author Response

Dear reviewer,

Thank you for your kindly comments on our manuscript. Please see below our response to your comments in the attachment. We appreciate your great efforts and constructive suggestions very much. If the manuscript still has some problems, please kindly point out for us and give us another opportunity to revise them.

Thank you again for your great help in improving the quality of the manuscript.

Yours faithfully,

Jun Yan

Qian Yao
